# Does a Little Difference Make a Big Difference? Bovine β-Casein A1 and A2 Variants and Human Health—An Update

**DOI:** 10.3390/ijms232415637

**Published:** 2022-12-09

**Authors:** Anna Cieślińska, Ewa Fiedorowicz, Dominika Rozmus, Edyta Sienkiewicz-Szłapka, Beata Jarmołowska, Stanisław Kamiński

**Affiliations:** 1Department of Biochemistry, Faculty of Biology and Biotechnology, University of Warmia and Mazury, 10-719 Olsztyn, Poland; 2Department of Animal Genetics, University of Warmia and Mazury, 10-719 Olsztyn, Poland

**Keywords:** bovine, β-casein, β-casomorphin-7, A2 milk

## Abstract

For over 20 years, bovine beta-casein has been a subject of increasing scientific interest because its genetic A1 variant during gastrointestinal digestion releases opioid-like peptide β-casomorphin-7 (β-CM-7). Since β-CM-7 is involved in the dysregulation of many physiological processes, there is a growing discussion of whether the consumption of the β-casein A1 variant has an influence on human health. In the last decade, the number of papers dealing with this problem has substantially increased. The newest clinical studies on humans showed a negative effect of variant A1 on serum glutathione level, digestive well-being, cognitive performance score in children, and mood score in women. Scientific reports in this field can affect the policies of dairy cattle breeders and the milk industry, leading to the elimination of allele A1 in dairy cattle populations and promoting milk products based on milk from cows with the A2A2 genotype. More scientific proof, especially in well-designed clinical studies, is necessary to determine whether a little difference in the β-casein amino acid sequence negatively affects the health of milk consumers.

## 1. Introduction

Milk accompanies humans from childhood, first as food from their mother, and then as a product used in the daily diet. It should be noted that humans are almost the only such case in nature who ingest milk in adulthood and who consume the milk of other species. From an evolutionary point of view, this phenomenon is relatively new, as historical data suggest that it has existed for about three thousand years. An increase in consumer interest in milk and dairy products has recently been observed [1]. One aspect of this interest is a discussion about the potential influence of the β-casein A1 variant on human health. The hypothesis assuming that the A1 variant of β-casein may have a negative effect on the health of milk consumers was formulated at the beginning of this century. According to this hypothesis, the digestion of cows’ milk with the A1 variant in the gastrointestinal tract may give rise to the opioid peptide β-casomorphin-7, which could be a risk factor in the development of heart disease, insulin-dependent diabetes, or neurological diseases. Unfortunately, there are many conflicting reports on this subject in the scientific literature. The numerous studies and reports that have appeared in recent years do not provide a definitive answer of whether milk containing the A1 variant of β-casein has a negative effect on the human body, and whether there are grounds for avoiding the A1 allele [2,3,4].

The significance and possible negative effects of the β-casein variant A1 on human health have been widely discussed and have aroused interest in the institutions in charge of food safety. The first such report was prepared for the Food Safety Authority of New Zealand [5]. This document concluded that the effects of β-casein A1 consumption on human health are important and require further investigation. Although the report of the European Food Safety Authority (EFSA 2009) “Scientific Report—Review of the potential health impact of β-casomorphins and related peptides” has not supported the hypothesis of the causal relationship between β-casomorphin-7 exposure and the etiology of human diseases, it also does not resolve the credibility hypothesis of the negative impact of the A1 β-casein variant, arguing that there is insufficient evidence and suggests further research in this field [6].

The hypothesis that a high consumption of A1 β-casein increases the risk of diabetes mellitus type 1 (DM-1), ischemic heart disease (IHD), sudden infant death syndrome (SIDS), schizophrenia, and autism spectrum disorder (ASD) is very intriguing and interesting for basic as well as application studies [2,3,4]. The topic is very vital and is of interest not only to science, but also to the breeding and dairy industry. Therefore, this article provides a summary and update of the existing knowledge and reviews focused on the β-casein gene and discusses the potential influence of β-casein variants on human health.

## 2. β-Casein Gene

### 2.1. Structure and Genetic Variants

β-casein (β-CN, 25–35% of milk proteins) is one of four caseins that together constitute nearly 80% of cow’s milk proteins [4,7,8]. Casein genes have been mapped in cattle on chromosome 6 (BTA6) in the region q31–33 and in the order: CSN1S1 (αs1-casein), CSN2 (β-casein), CSN1S2 (αs2-casein), CSN3 (κ-casein), starting from the 5’ end [9,10,11,12]. Casein cluster is represented by a DNA fragment of about 250 kb [9,11,13] and its expression is coordinated by multihormonal factors. This region is very conservative in many mammalian species, both in terms of construction and organization [13].

The first complete sequence of the bovine β-casein gene (CSN2, 8.5 kb) was published by Bonsing et al. [14] in 1998. Nowadays, we know that this gene consists of nine exons and eight introns, has a total length of 10,338 bp (GenBank: M55158.1), and is classified as the most polymorphic gene of all bovine caseins, with most mutations located in exon 7. The polymorphism of β-casein was first discovered by Aschaffenburg in 1961 [15]. Up until now, 15 genetic variants of β-casein coding regions have been reported and named according to the order of discovery as A1, A2 [16,17], A3 [17], B, C [18,19], D [15], E [20], F [21], G [22], H1 [23], H2 [24], I [25], J, K, L [26]. For the next two reported variants of bovine β-CN, A4 and B2, nucleotide substitutions have not been recognized yet. An additional A5 variant with nucleotide substitution found in the intron of the β-casein gene had no implication on the protein structure. All identified and confirmed changes in the amino acid sequence of the β-casein variants are shown in Figure 1. Since the primary gene product (224 amino acids; GenBank: AAA30431.1) also contains a signal peptide that is removed post-transcriptionally, the final protein molecule is composed of 209 amino acid residues [27,28]. Basically, all the β-casein variants differed in 1–3 amino acid substitutions at different positions, but they could generally be classified as the β-casein A2 type (10 variants) or A1 type (5 variants) depending on the Pro or His presence (respectively) at position 67 of the protein sequence. The cause of it is a single nucleotide polymorphism (SNP) at codon 67 of the β-casein gene in exon 7. It is accepted that SNP67 is the effect of the natural mutation with the change of cytosine (A2 allele: CCT, proline) into adenine (A1 allele: CAT, histidine) [29,30].

### 2.2. Variants and Their Frequency in Dairy Cattle Breeds

Many early data on the presence and prevalence of β-CN variants were based on a starch gel electrophoresis, and the method that allowed for the differentiation of only A, B, and C variants [18,19]. Later, Seibert et al. in 1985 [31] and Caroli et al. in 2016 [32] proposed the isoelectric focusing electrophoresis (IEF) method for the detection of the β-casein A1, A2, A3, B, and C variants in bovine milk. At the same time, chromatographic methods (RP-HPLC) and mass spectrometry (MS) have been proposed for the identification of variant F, the β-casein with the electroneutral substitution of amino acid with a different hydrophobic index [21]. Generally, genetic methods such as PCR-RFLP, Real-Time PCR, or sequencing are used to identify polymorphism in the β-casein gene.

The most common variants of β-casein in dairy cattle breeds are A1 and A2, while B is less common, and A3 and C are considered rare [33]. Other β-CN types are very rare or identified exclusively in humped cattle (Zebu) or African cattle (Ankole) [15,20,34,35,36] [15,20,34,35,36]. Only seven of β-CN variants (A1, A2, A3, B, C, I, and E) have been detected in European cattle breeds. Although reported as the second type, the A2 variant is considered as the oldest one from which the others originated via mutation [37]. The presence of this variant is dominant in African and Asian cattle, where the average A1 and A2 frequency was found at the level of 0.16 and 0.82, respectively [15,20,34,35,38]. Studies conducted over the last few decades within European, American, and Australian dairy cattle populations indicate that the average prevalence of the A1 allele in these regions is 0.35, while the A2 allele is 0.61. The distribution of frequency clearly depends on the breed of cattle. Variant A1 is dominant in Ayrshire and Red Denmark cattle (between 0.51 and 0.72), while its low frequency is found in Guernsey, Jersey, Brown Swiss, and Brown Italian cows (between 0.04 and 0.14). In the world’s most common breed of dairy cattle, Holstein-Friesian (HF), the frequency of the A1 variant ranges from 0.25 to 0.51. The occurrence of the CSN2 A1 allele in various dairy breeds and countries is shown in Figure 2 (based on [22,39,40,41,42,43,44,45,46,47,48,49,50,51,52,53,54,55,56,57,58,59,60]).

## 3. Beta-Casein Protein

The biological function of the αs1-CN, αs2-CN, and β-CN is to form micelles, which are macromolecular structures responsible for the transfer of calcium to the newborn. The fourth casein (κ-CN) is the stabilizing factor of the micelles and could play an important protective function against *Helicobacter pylori* infections in infants [11,61].

β-casein is the most hydrophobic casein multilaterally bounded in casein micelles in an aqueous solution. As it lacks Cys, β-casein has a flexible and open conformation with a little tertiary structure. The C-terminal region of β-casein is hydrophobic, while the N-terminal domain rich in phosphate groups is highly negatively charged and polar [62]. Phosphoserine residues in the polar part of the molecule interacts electrostatically with colloidal calcium phosphate (CCP) to form nanoclusters. The nonpolar part of the molecule enhances micellar stability by forming hydrophobic bonds with other caseins [62,63,64].

A study by Raynes et al. showed structural differences between bovine A1 and A2 β-casein [65]. As already mentioned, β-casein participates in the formation of the casein micelles, forms oligomeric micelles itself, and functions as a molecular chaperone, preventing the aggregation of a wide range of proteins that also include other caseins. Differences in micelle assembly and chaperone activity may explain differences in the functionality of A1 and A2 milk. The A2 β-casein variant forms smaller micelles than the A1 β-casein. The monomer-micelle form equilibrium of the A2 β-casein is shifted toward the monomer, where the shift comes from the structural differences between the two β-casein variants associated with the adoption of the greater polyproline-II helix in the A2 β-casein, which may lead to enhanced chaperone activity of the A2 β-casein in comparison to the A1 β-casein [65]. To our best knowledge, the first study investigating the differences in the protein composition of casein micelles, milk whey, and fat globule membrane (MFGM) in three milk variants (A1A1, A2A2, and A1A2) was presented by Wang et al. (2020), who used the proteomic method with a label-free approach to analyze this aspect [66]. They found different contents of the protein cargo not only in casein micelles, but also in the whey and MFGM fractions. The overall analysis of these three fractions showed that several proteins were significantly associated with each of the milk variants including ceruloplasmin, protein S100-A9, and cathelicidin-2 in A1A1 milk, lactoferrin, protein S100-A8, CD5L, and protein S100-A12 in A2A2 milk, and selenoprotein P, β-glucuronidase, and osteopontin in A1A2 milk. However, the genetic rationale for these findings corresponding to the β-casein variants, its biological meaning, and physiological implications for consumer health remains unclear.

## 4. β-Casein Digestion

β-casein is a relatively slowly digestible protein that can be completely degraded, or due to the limited proteolysis, can release bioactive peptides with antioxidant, ACE-inhibitory, or opioid activity in a way that is dependent on the genetic polymorphism [67,68,69,70]. Especially interesting here are morphine-like peptides (β-casomorphins). They are suspected to play an important role in the response to stress, pain, regulation of food intake, or perform other pathobiological functions as they are able to bind to the opioid μ-receptors (MORs) found principally in the central nervous system, immunological system, and the gastrointestinal tract [71,72,73,74,75]. Thus, as milk and dairy products are often the main component of the diet, their consumption may predispose hypersensitive individuals to adverse health effects. Peptides can be released from the parent structure in a few ways: hydrolysis by digestive enzymes in the digestive tract, as a consequence of microbiome activity, or as an effect of technological food processing [76,77,78,79,80,81,82,83,84,85].

## 5. β-Casomorphins

β-casomorphins (βCMs) are a group of peptides with a chain length of 4–11 amino acids, all starting with the tyrosine residue critical to their opioid activity (Figure 3) [70]. The first isolated, and the most often identified later on, was the β-casomorphin-7 (BCM-7) heptapeptide, the sequence of which corresponds to the fragment 60–66 of the parent protein [68]. It was shown that in simulated gastrointestinal conditions in vitro, β-CM-7 is yielded by the successive gastrointestinal proteolytic digestion of β-casein A1 and B (but not A2) by pepsin, pancreatic elastase, and leucine aminopeptidase [82,84,86]. The cause of this difference is due to single nucleotide polymorphism of the β-casein gene (SNP67) and proline substitution by histidine in A1 of the β-casein molecule. This amino acid substitution results in the conformational difference in the expressed protein secondary structure, which may exert an influence on the physical properties of the respective casein micelles [87,88]. Additionally, the peptide bond between proline and isoleucine in the A2 variant has higher enzymatic resistance than that between histidine and isoleucine in the A1 variant. Therefore, the A1 β-casein is more readily hydrolyzed, resulting in the release of β-CM-7 [84,89]. The release of β-CM-7 during simulated gastrointestinal digestion (SGID) of A1A1 and A1A2 milk β-casein was confirmed in vitro [52]. However, it should be noted that Cieślińska et al. [53] and Duarte-Vazquez et al. [90] further showed that small amounts of β-casomorphin-7 could also be produced from β-casein A2. The release of β-casomorphin-7 from both the A1 and A2 milk β-caseins was recently confirmed by Lambers et al. [91], but not by Haq et al. [86], who did not find β-CMs in the hydrolyzed milk A2. The presence of β-casomorphin-7 was also identified in vivo, in the jejunum of healthy humans who ingested bovine milk or casein. Although the authors did not specify the parental protein variant, they estimated that the amount of β-casomorphin-7 was sufficient to elicit its biological action [92,93,94].

Due to the high-proline structure, β-CMs are very stable regarding enzymatic degradation by most peptidases and proteinases. They also were found to be resistant to microbial aminopeptidases [96]. Asledottir et al. [97] studied β-CM-7 degradation and stability and used human gastrointestinal juice and porcine jejunal brush border membrane (BBM) peptidases. Products were next profiled using HPLC-electrospray ionization mass spectrometry (ESI/MS) to monitor β-CM-7 during the gastrointestinal digestion process. Intact β-CM-7 was quantified using RP-HPLC. The experiment showed that β-CM-7 is partly digested with gastrointestinal enzymes. Aside from the detection of three different proteolytic fragments (f(62–66) FPGPI, f(60–65) YPFPGP, and f(61–66) PFPGPI), the entire peptide molecule f(60–66) YPFPGPI was also found. After 2 h of BBM digestion, it was reported that 42% of the initial peptide was degraded, and after 4 h, the results showed a degradation of 79%. However, a small amount of approximately 5% was still detectable after 24 h of gastrointestinal and BBM digestion. Generally, β-CMs are good substrates for only several enzymes. One of them is dipeptidyl-peptidase IV (DPP4, CD26), which is a cell-surface protease belonging to the prolyl oligopeptidase family. DPP4 is expressed on epithelial cells, immune system cells, and is present in a soluble form in the blood and extracellular fluids [98,99]. Kreil et al. found that plasma DPP4 hydrolyzes β-CM-5 to a mixture of YP, FPG, FP, and G [100], whereas Osborne et al. showed the rapid hydrolysis of β-CM-7 by the model of the intestinal epithelium (Caco-2 cells) with the production of three peptide metabolites: YP, GPI, and FPGPI [101].

Enzymatic resistance is probably one of the most important factors related to BCM bioavailability. Sienkiewicz-Szłapka et al. [102] have reported that at least two β-casomorphins, β-CM-5 and β-CM-7, are capable of crossing the intestinal epithelial cell monolayer. Transportation of both peptides was found with a low permeability rate in the presence of the full DPP4 activity, whereas inhibition of this enzyme activity increased the β-casomorphin absorption, even ten times in the case of β-CM-7. Moreover, Jarmołowska et al. [103] indicated that the transport of intact β-CM-7 could be determined not only by brush border hydrolase activity, but also by food ingredients. They found that peptide transport efficiency was enhanced by increased glucose and calcium levels in the culture medium. This strongly suggests that disturbances to the intestine barrier (“leaky gut”) and a high-sugar modern diet could promote food-derived opioid peptide penetration. On the other hand, it also provides a possible explanation for the lack of β-CM-7 in the blood and urine following the consumption of casein in healthy adult persons with normal functionality of the intestine barrier [101].

## 6. β-Casomorphin-7 in Milk and Milk Products

β-Casomorphins and their precursors have been identified in milk and various dairy products. A quantitative examination of the β-CM-7 in the fresh and hydrolyzed (by digestive enzymes) bovine milk revealed that in hydrolyzed A1 milk, there was a 4-fold higher level of β-CM-7 than in A2 milk, whereas in the non-hydrolyzed milk, traces of β-CM-7 were found [52,89]. Small amounts of β-CM-7 after digestion of the A2 milk β-casein were also detected by Duarte-Vazquez et al. [90] and Lambers et al. [91]. Other results were obtained by Haq et al. [86], who found a 3.2 times higher level of β-CM-7 released from the A1A1 variant after enzymatic digestion in comparison to the A1A2 variant of β-casein, and no β-casomorphin-7 after the digestion of the A2A2 variant of β-casein. It should be noted here that Lambers et al. also found that higher amounts of this peptide were liberated from the raw milk proteins than from heat-processed milk [91]. However, these results were not confirmed by our research (unpublished data).

β-Casein-derived opioid peptides have been identified in fermented milk products and different types of cheeses. An example of fermented milk drinks in which β-casomorphin-7 has been identified are natural yogurt and kefir [104,105]. The content of peptides was rather low in the examined products, but as suggested by Nguyen et al., factors such as the time of fermentation, time, and conditions of product storage could strongly influence the opioid peptide concentration [106]. Precursors of β-CMs or β-CM-9 and β-CM-10 were also found in Gouda, Swiss, Blue, Limburger, and Brie cheeses, but not in mature Cheddar cheese, perhaps due to degradation during the ripening process [107,108,109]. Other researchers have reported the presence of β-casomorphin-7 in Gorgonzola, Gouda, Fontina, and Cheddar [104], Edamski, Gouda, Kasztelan, Rokpol and Brie, Kaszkawał, and Camping and Brie cheese [110,111]. Many of these findings were qualitative, however, based on the available data, it seems that short-ripening soft cheeses (mold-cheeses, French type) contain more β-CM-7 than the Dutch-type semi-hard cheeses that are riper for longer.

Finally, several reports also showed the presence of β-CM-like and morphiceptin-like activities or exactly β-casomorphin-5 and -7 in infant formulas [104,112,113]. Working in this area, Duarte-Vazquez et al. developed an infant formula based on β-casein A2 milk where the concentration of β-CM7 was significantly lower than in other tested infant formulas including a formula based on A1 β-casein milk [90].

## 7. β-Casein Variants A1/A2 and Human Health

As presented above, it is thought that the β-casein variant A1 yields the bioactive peptide β-casomorphin-7, which is thought to play a role in the higher incidence of some human diseases. At the end of the 1990s, some reports suggested that casein variant A1 consumption is a risk factor of type 1 (insulin-dependent) diabetes mellitus [114] and ischemic heart disease in humans [115]. Additionally, a relation of β-casomorphin to sudden infant death syndrome (SIDS) [2,87,88,116,117,118,119] and autism [120,121] has been suggested. Another potential impact of milk proteins on human health is its hypothetical correlation to milk allergy and atopic dermatitis (AD) [122,123,124,125]. In contrast, Zoghbi et al. claim that dairy products containing β-casomorphin-7 may improve intestinal protection and could have dietary and health applications [126]. β-CM-7 is known to influence the endocrine, nervous, and immune systems by activating μ-opioid receptors, which leads to different effects such as analgesia, sedation, reduced blood pressure, nausea, decreasing respiration, and bowel motility [114]. The known influence of β-CM-7 on human body systems is presented in Figure 4.

### 7.1. Lactose Intolerance

Digestive disorders after milk consumption are correlated not only with lactose intolerance, but also with the type of casein found in the consumed milk. Milan et al. conducted a study on a group of 40 women with self-reported varying dairy tolerance [127]. Participants ingested 750 mL of conventional milk (with A1 and A2 β-casein and lactose), A2 milk (with A2 β-casein and lactose), or lactose-free conventional milk (with A1 and A2 β-casein, but without lactose). In patients diagnosed with lactose intolerance, consuming A2 milk brought significant health benefits compared to consuming conventional milk, for example, reduced intensity (*p* < 0.05) of nausea, fecal urgency, and the rise in breath hydrogen. On the other hand, this mechanism has not been confirmed in patients with non-lactose dairy intolerance who manifested symptoms such as abdominal distension, bloating, and flatulence, but not increased breath hydrogen [128]. A similar study was performed by Ramakrishnan et al. [129], who concluded that intaking A2 milk caused fewer symptoms of lactose intolerance than conventional milk (with both, A1 and A2 β-caseins) in patients with lactose intolerance and/or digestive problems. The authors showed that the consumption of milk containing A2 significantly enhanced symptom scores (*p* = 0.04) and reduced hydrogen production (*p* = 0.04). The obtained results were the effect of a randomized, double-blind, crossover trial (a single-meal study), which was carried out on 25 participants (Ramakrishnan et al. 2020). A greater number of participants were included in the analysis performed by He et al. [127], who conducted a study with 600 patients randomly assigned to drink A1/A2 β-casein milk or A2 β-casein milk. Participants reported symptoms using 9-point visual analogue scales for gastrointestinal symptoms, and the following parameters were tested in the study: borborygmus, flatulence, bloating, abdominal pain, stool frequency, and stool consistency. Symptom rates were significantly lower after intaking A2 milk, with simultaneous relieving acute gastrointestinal symptoms. The authors also noted that conventional milk containing β A1-casein reduced the activity of lactase and intensified dysfunctions in the digestive system. Therefore, one can conclude that in some individuals, gastrointestinal symptoms related to milk may be due to the ingestion of A1 β-casein, and this mechanism is not related to lactose intolerance [127].

The effects of consuming A2 milk in correlation with the opioid system have also been described. Using in vivo models, it was proven that A1 milk consumption delays intestinal transit in rodents compared to A2 milk intake, and it is accompanied by an opioid-dependent mechanism [130]. The use of animal models also allowed us to conclude that the consumption of A1 milk causes the initiation of inflammatory markers and an increased expression of the Toll-like receptors, which play a crucial role in the innate immune system. There is also evidence that A1 milk consumption is associated with delayed intestinal transit, looser stool consistency, digestive discomfort, and the synthesis of inflammatory markers in humans, but this approach still requires further research [130,131]. Described indications support the increasing availability and promotion of A2 milk, which can bring great benefits in improving the quality of life of infants, children, and adults.

### 7.2. Inflammatory Response and Gastrointestinal Problems

β-Casomorphins and other peptides released from β-casein may also affect the human mucosal immune [132,133,134] via opioid receptors, thus may serve an important function in the digestive process. Fiedorowicz et al. [135] investigated the effect of infant formula hydrolysates on the functioning of the intestinal epithelium in a model Caco-2 cell system. They showed that hydrolysates containing β-CM-7 resulted in the significant alteration of intestinal epithelial cell proliferation and increased the secretion of IL-8. Modulation of this potent neutrophil chemoattractant secretion may have an influence on chemotaxis, degranulation, and the production of superoxide anions in the process of phagocytosis in these cells [135,136]. Moreover, they showed that the tested hydrolysates stimulated the adhesion of *Bifidobacterium* and reduced the adhesion of *Enterobacteria* isolated from the infants’ intestines to the epithelial cells, which may also affect the balance in the intestinal microbiota.

Pal et al. [137] suggest that bovine β-casomorphin-7, derived from A1 β-casein, could also be an important contributor to milk intolerance syndrome. Haq et al. [138] and Barnett et al. [139] confirmed that the consumption of β-casomorphin-5 and -7 induced an inflammatory immune response in the guts of mice and rats, respectively. Moreover, human trials performed by Ho et al. [140] suggest differences in gastrointestinal responses in some adult humans consuming milk containing A1 or A2 β-casein. In the tests, A1 β-casein milk led to significantly higher stool consistency and a significant positive association between abdominal pain and stool consistency values compared with the A2 β-casein milk. A study conducted by Jianqin et al. [141] also supported the thesis that A1A1 (and A1A2) β-casein consumption by subjects with lactose intolerance was associated with gastrointestinal inflammation, and negative post-dairy digestive discomfort symptoms. Pseudo-allergic skin reactions to opiate sequences of bovine casein in healthy children were presented by Pal et al. [137], Brooke-Taylor et al. [130], and Cieślińska et al. [142]. Deth et al. [143] demonstrated in the human study that the consumption of milk containing only A2 β-casein was associated with a greater increase in plasma glutathione concentrations compared with the consumption of milk containing both β-casein types. The hypothesis is in line with Sheng et al. [144], who compared the effect of 5 days of consumption of conventional milk to the effect of only A2-β-casein milk. People who consumed milk containing only A2-β-casein had significantly fewer gastrointestinal symptoms, reduced stool frequency, and improved stool consistency. Effects were observed in preschoolers with mild-to-moderate milk intolerance. Results contained measurements including HGB, IL-4, IgG, IgG1, β-CM-7, GSH, CRP, and IgE. It was found that even short-term consumption of conventional milk significantly increased both proinflammatory markers related to the Th2 response and β-CM-7 level in the serum (*p* < 0.0001), while in the case of A2 milk consumption, all of the parameters showed no statistically significant difference. Based on the results, the authors concluded that the exclusion of A1 milk from the diet may help alleviate adverse gastrointestinal symptoms in lactose-intolerant children [144].

### 7.3. Allergy

It is suggested that β-casomorphins can induce pseudo-allergic reactions by histamine release from immune cells. Stepnik and Kurek [145] showed inductive effects on the mast cells of rodents after preincubation with β-CM-7. Immediate and dose-dependent wheal and flare reactions in the skin of healthy children were observed after β-casomorphin-7 intradermal injection. These skin reactions were inhibited by cetirizine. On the other hand, Fiedorowicz et al. [146] showed that incubation of PBMCs with peptide extracts from cow A1 milk caused an increase in the μ-opioid receptor (MOR) gene expression in children with atopic dermatitis with a simultaneous decrease in DPP4 gene expression. Moreover, Fiedorowicz et al. [147] detected the changes in the secretion of IL-4 and IL-13 by the PBMCs under the influence of β-casomorphin-7. It may provide immunomodulatory effects of this opioid peptide and the role in allergic diseases, as produced by CD4+ T lymphocytes IL-4, acts like a β-lymphocytes growth and differentiation factor, and stimulates the production of IgE [136,147].

### 7.4. Diabetes Mellitus

Type I diabetes (DM1) is a disorder characterized by autoimmune destruction of the pancreatic cells by T cells and macrophages [148] whereby the organism loses its ability to produce insulin. The biological mechanism of this disease is unknown [149]. This disease occurs in children under 14 years of age. Cavallo et al. [150] showed that antibodies against β-casein increased in DM-1, and suggested the hypothesis that casein may play a role in that disease. This was supported by Elliott et al. [87], who demonstrated, in an animal model, that the A1 fraction of β-casein was found to be diabetogenic for non-obese diabetic (NOD) mice, whereas the A2 fraction was not. Moreover, epidemiological studies have shown a significant association between the intake of A1 milk (but not A2 milk) and the incidence of DM-1 [88,118]. Elliott et al. [87] compared DM-1 incidence in 0–14-year-old children from 10 countries with the national annual cow milk protein consumption. He showed that in Iceland, where cows are predominantly A2, there are low numbers of cases of diabetes and heart disease. He noted that the distinctive peptide that formed mostly from the A1 β-casein and partly from B β-casein was β-CM-7, and that this was a hypothetical risk factor of the disease. Data presented by Thorsdottir et al. [116] and Birgisdottir et al. [151] supported this hypothesis. It was shown that in Finland, Norway, Denmark, and Sweden, where the cows are mainly A1, that the incidence of the disease was very high. It is important to note in these studies that these are countries where the external factors—climate, diet, and population of people—are the same or very similar, and the only differentiating factor was the milk from cows with alternative genotypes of β-casein. Additionally, McLachlan [88] showed that the consumption of the A1 β-casein across 16 countries was strongly correlated with the incidence of DM-1 in children under 15 years of age. In 2003, Laugesen and Elliott [117] confirmed the same relationship between A1 and DM-1 across 19 countries in 0–14-year-old children. Correlations were not significant for the variant A2 of β-casein.

Several mechanisms have been suggested to explain the risk of developing insulin-dependent diabetes associated with consumption of the β-casein A1. It is possible that opioid peptide β-CM-7, produced from β-CN A1, influences the development of gut-associated immune tolerance, the immune surveillance, or suppresses the defense mechanisms toward enteroviruses or endogenous retroviruses, which then damage the pancreatic β-cells [87,117,150,152]. Another explanation has been found by Monetini et al. [153]. Based on studies on the cross-reactivity of T lymphocytes from DM-1 patients exposed to β-CN, they proposed a mechanism based on BCN protein molecular mimicry to the epitope on the GLUT-2 (glucose transporter 2) of pancreatic β-cells. According to this hypothesis, molecular mimicry of the β-CM-7 peptide fragment (corresponding to f63-67 β-CN) can inhibit GLUT-2 activity by generating autoantibodies interacting with GLUT-2 epitopes (f415-419) on β-cells. These results were confirmed in the studies by Chia et al. [154]. Other possible links between β-CN A1 and diabetes emerged with the demonstration that β-CM-7 is capable of modulating cysteine uptake in cultured human neuronal and gastrointestinal (GI) epithelial cells via the activation of opioid receptors. β-CM-7 decreases cysteine uptake by excitatory amino acid transporter 3 (EAAT3) and results in the decrease in the intracellular antioxidant glutathione pool (GSSG/GSH). In the author’s opinion, restricted antioxidant capacity, caused by milk-derived opioid peptides, may predispose susceptible individuals to inflammation and systemic oxidation stress development [155]. As indicated by Bruni et al. [156] and Kay et al. [131], β-CM-7 decreases the GSH concentrations, which could contribute to the ferroptosis of pancreatic β cells and as a consequence of the programmed cell death resulting from iron-dependent lipid peroxidation accumulation. Administering an opioid receptor inhibitor was proven to partially reverse this effect [131,154]. Finally, β-CM-7 can cause the dysregulation of insulin action and disruption of metabolic processes, affecting the glucose regulation mechanisms directly via the opioid mechanism. Activation of μ-opioid receptors on the membranes of insulin target cells could inhibit the phosphoinositide 3-kinase (PI3K) signaling pathway and disturb GLUT-4 (glucose transporter 4) vesicle transportation to the cell membrane. All of the above-mentioned possible mechanisms of the participation of β-CN A1/β-CM-7 in the etiopathology of diabetes are presented in Figure 5. A wide review of epidemiological, animal-based, in vitro, and theoretical evidence for A1 β-casein as the dominant causal trigger of type 1 diabetes has been presented by Chia et al. [154].

On the other hand, however, it has to be noted that the experiments of Yin et al. [157] showed the opposite, protective effect of β-casomorphin-7 on type 1 diabetes rats, where the peptide was found to be a reducer of the elevated blood glucose level. Furthermore, it was suggested that the administration of β-casomorphin-7 reduces the absorption of glucose and thus decreases the high glucose-induced oxidative stress [158,159] and attenuates renal interstitial fibrosis caused by diabetes [160]. Furthermore, in the opinion of Clemens [161], the evidence from several epidemiological studies and animal models does not support the association between milk proteins and the development of diabetes in children.

### 7.5. Ischemic Heart Disease

Correlations between A1 β-casein consumption and human disease are mainly based on epidemiological studies [2] also regarding the circulatory system. Epidemiological evidence from New Zealand suggests that consumption of A1 milk could be associated with heart disease incidence for 30–69-year-old males [88]. Based on the performed analyses, he postulated that β-casein A1, or possibly its fragment (peptide BCM-7), could be a significant contributor to the etiology of cardiovascular disease in this population. Additionally, Laugesen and Elliott [117], in retrospective studies over the course of 15 years (1980–1995) and in 20 affluent countries around the world, showed a relationship between A1 β-casein and IHD mortality in 35–64-year-old males. Several experimental studies have provided probable mechanisms explaining or contradicting the observed correlations. McLachlan [88] suggests here the role of β-casein consumption in the development of hypercholesterolemia or atherosclerosis in numerous animal studies including rabbits, pigs, monkeys, and rodents. Additionally, rabbits fed with β-casein A1 milk had higher cholesterol levels and higher percent surface area of the aorta covered by fatty streaks than those fed with β-casein A2 [119]. Torreilles and Guerin [162] and Steinerova et al. [163,164] [165,166] showed that β-casomorphin-7 derived from β-casein A1 promoted the oxidation of human LDL, which correlated with an increased risk of heart disease. In humans, the content of oxidized LDL is considered the first step in the development of atherosclerosis [88]. On the other hand, A2 β-casein consumption can protect against IHD as the low-density lipoprotein (LDL) and high-density lipoprotein (HDL) cholesterol levels were lower on the A2 diet than on the A1 diet. The physiological effect of β-CM-7 in A1 milk on the oxidation of LDL or peroxidation of a lipid component of LDL, regarded as a determining step in the development of heart disease, has been shown [87]. The analysis of protein oxidation products isolated from atherosclerotic lesions implicates the tyrosyl radical [165], and β-CM-7 is its potential source. In contrast, the research of Kamiński et al. [166] showed that there was no relationship between the A1 β-casein or A1 milk feeding of piglets and the abnormality in basic hematological and biochemical indices. Additionally, human trials did not show any significant differences in the blood parameters (triglycerols, total cholesterol, HDL, LDL cholesterol) in groups that consumed A1 or A2 β-casein milk [167]. Moreover, Dong-Ning Han et al. [168] even showed the protective effect of β-casomorphin-7 on cardiomyopathy in rats. Petrat-Melin et al. [67] indicated that the hydrolysate A1 and B variants of β-casein increased 5-fold angiotensin-converting enzyme ACE inhibition in vitro in comparison to the A2 and I variant hydrolysates (3.0-fold). This indicates that A1 milk has a more hypotensive effect than A2 milk, and thus could potentially be more beneficial for patients with ischemic heart disease, heart failure, and impaired contractility of the left ventricle.

### 7.6. Nervous System

It is hypothesized that similarly to pharmacological opioids, food-derived exorphins such as β-CM-7 can also cross the blood–brain barrier (BBB). It is likely that they can bind to transporting proteins that protect them from the hydrolytic action of peptidases in the blood and then could become a substrate for the carrier peptide transport system-1 (PTS-1) in the BBB [118,136,169,170,171]. As β-CM-7 is a μ-opioid receptor (MOR) agonist when it crosses the BBB, this could activate the respective receptors of CNS, a crucial component of the internal messaging systems that involve endorphins and enkephalins. This would lead to neurological disorders or altered neuronal development [172]. However, opioid receptors can be found not only in the CNS, but they are also present in the peripheral nervous system, immune and endocrine system, and even on the bone cells. Thus, the opioid receptors present across many parts of the human body allow opioids to exert their functional effects from pain management to bone metabolism regulation.

The consumption of β-casein A1 could be linked to some neurological, neurodevelopmental, and mental disorders. Sun and Cade [173] showed that β-casomorphin-7 is able to accumulate in the rat’s brain regions relevant to schizophrenia and autism. A few years later, the role of opioid peptides in human autism etiology has been suggested by Reichelt et al. [174], whereas Sokolov et al. [175], based on clinical observations and laboratory results, stated that chronic exposure to elevated levels of bovine β-casomorphins may impair early child development, setting the stage for autistic disorders. Significantly higher levels of β-CM-7 in the urine [174,176] and blood [120] of patients with schizophrenia, autism, and women with postpartum psychosis has already been proven several times, whereas in the in vitro tests, it was found that β-casomorphin-7 may influence the expression of its receptor gene in autistic children [177]. Additionally, although opioid receptor activation was considered here to be a major mechanism, it was proven that β-CM-7 also reacts as an antagonist with 5-HT2 serotonin receptors [173,175]. This significantly extends the range of its biological activity.

SIDS is the cause of the death of infants between the end of the first month and the first year of life [130]. Sun et al. [118] indicated that one factor in common with all children who developed SIDS was milk—their only source of food. β-Casomorphins were detected in the plasma of infants fed cow’s milk infant formulae [120]. In neonates, the β-casomorphins are passively transported across the intestinal mucosal membranes [178] and following absorption, crossing of the blood–brain barrier can occur because of the infant’s immature systems [179]. In infants with abnormal respiratory control and vagal nerve development, the opioid peptides derived from milk might induce depression of the brain-stem respiratory centers, leading to death. It has been reported that β-CM immunoreactivity was found in the brain stem of human infants [180]. Wasilewska et al. [181,182] determined a higher level of β-casomorphin-5 and -7 and a lower activity of DPP-4 in the sera of infants with apparent life-threatening events (ALTE syndromes, ‘near miss SIDS’). These findings suggest that the lower activity of peptidase (prone infants) may be responsible for opioid-induced respiratory depression, induced by β-CM-7.

## 8. β-Casein Variants A1/A2 in Dairy Cattle Breeding

If the risks associated with variant A1 β-casein consumption are confirmed, consumers may wish to reduce or remove this kind of milk from their diet. The farmers should take appropriate steps to allow for a systematic reduction in the number of cows and bulls with the A1 allele of β-casein and consequently reduce the spread of this undesirable allele in a dairy cattle population. Genetic polymorphism related to the differences in animal breeding value can be considered in the selection process. Research by Oleński et al. [183,184] showed that the A1 allele is associated with lower levels of milk yield traits, and the A2 variant increased the breeding values for the milk yield and milk protein content. Norwegian researchers [184] have suggested increasing the frequency of the allele A2 β-casein in the Norwegian cattle population due to its very positive effect on milk traits. Similar conclusions were proposed by Heck et al. [49]. The work of Gustavsson et al. [56] suggests that a higher frequency of β-casein A1A2 could have positive effects on the processing of cheese. Additional benefits of the A2 variant have also been spotted and economically estimated by Morris et al. [185], who indicated that the A2A2 milk, because of the better characteristics, had a higher daily yield of milk (about 2.1% higher than the value of A1A1 and A1A2 together). Furthermore, Kearney et al. [186] calculated that A2A2 cows produce a higher profit for milk than A1A2 or A1A1 cows.

## 9. Conclusions

Summing up, although the hypothesis on the influence of the β-casein A1 variant on human health was expressed almost 20 years ago, it still lacks conclusive evidence for its confirmation. The established effects of A1 milk compared to A2 milk showed varied results on human health. While it seems that studies have reported that A2 milk has been characterized by significantly greater health benefits due to the fact of the increased supply of β-CM7 in A1 milk, often contradictory information coming from studies in animals and human trials suggests further research in this area. Nevertheless, the elimination of the A1 allele in the population of dairy cattle will not adversely affect the milk yield and milk composition. Scientific reports in this field can affect the policy of dairy cattle breeders and the milk industry, leading to the elimination of allele A1 in dairy cattle populations and promoting milk products based on milk from cows with the A2A2 genotype. More scientific proof, especially in well-designed clinical studies, is necessary to find whether a slight difference in the β-casein amino acid sequence negatively affects the health of milk consumers.

## Figures and Tables

**Figure 1 ijms-23-15637-f001:**
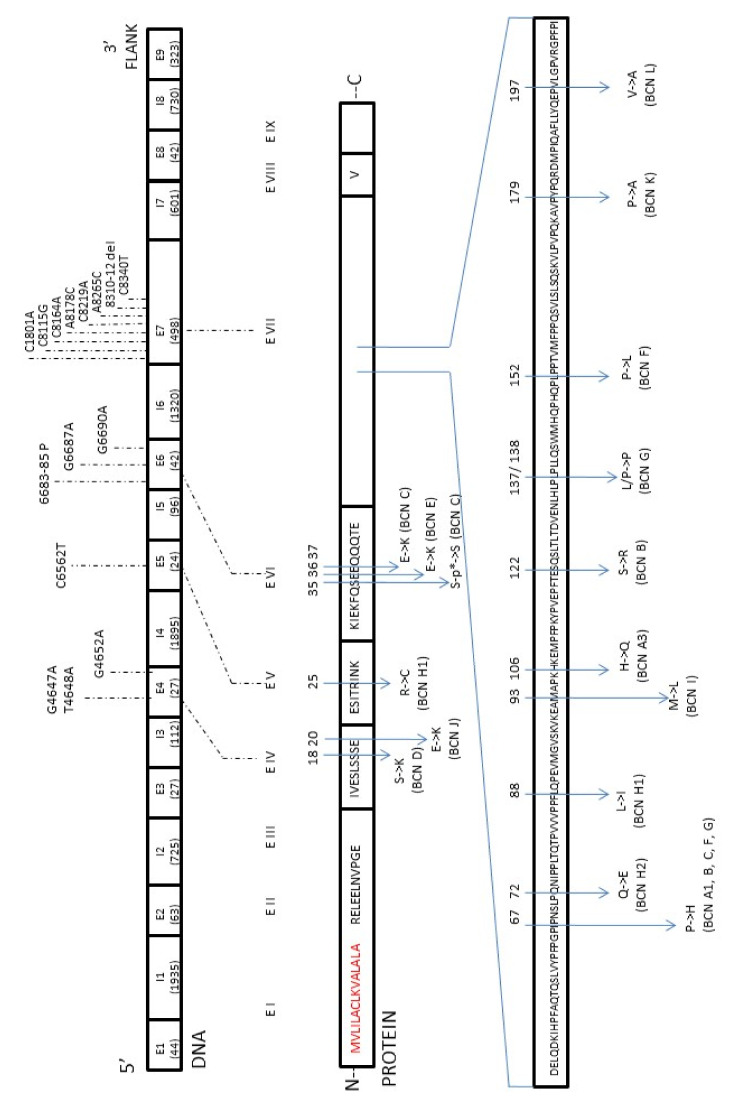
The map of mutations found in the β-casein (CSN2) gene in relation to the amino acid changes with the β-casein protein (β-CN).

**Figure 2 ijms-23-15637-f002:**
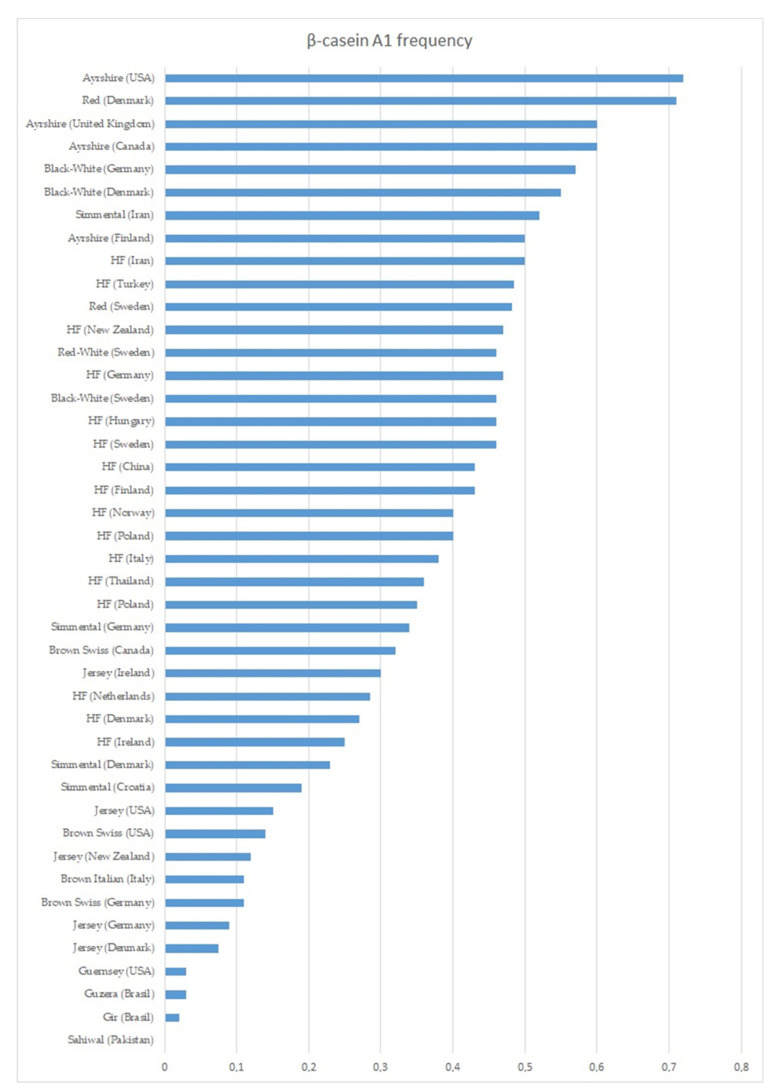
Frequency of the A1 β-casein gene variant in various breeds and countries.

**Figure 3 ijms-23-15637-f003:**
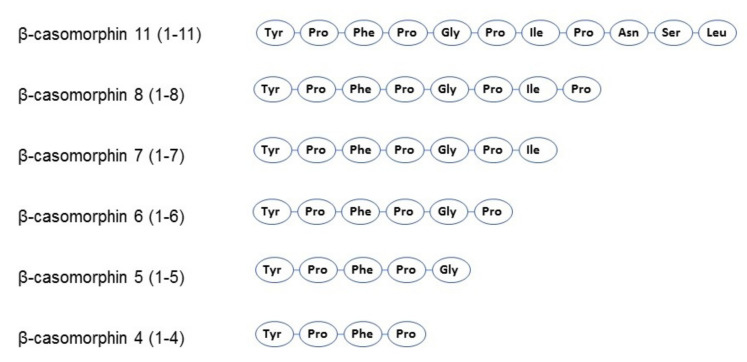
The amino acid composition of the β-casomorphins formed in cow’s milk (based on Ramabadran and Bansinath [95]).

**Figure 4 ijms-23-15637-f004:**
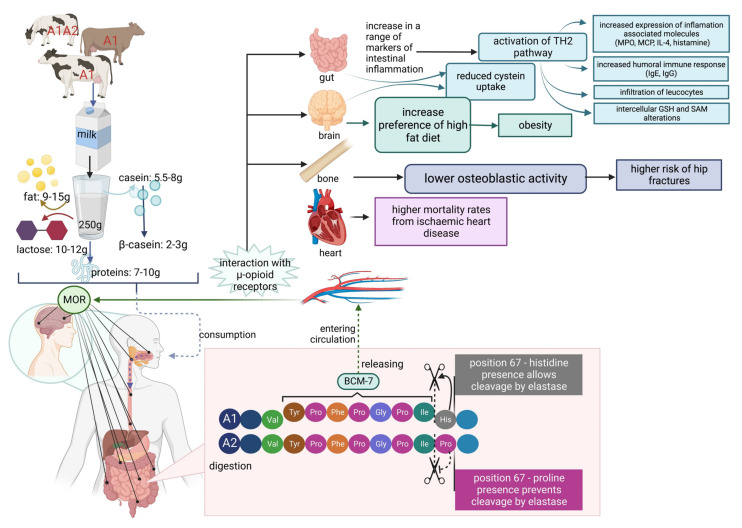
β-Casomorphin-7 influence on human body systems.

**Figure 5 ijms-23-15637-f005:**
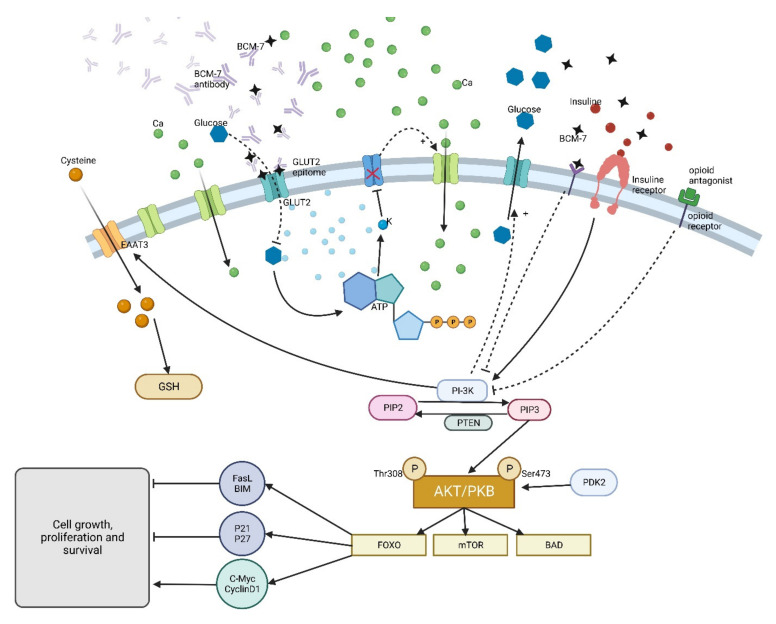
β-Casomorphin 7 pathway in the human cell.

## Data Availability

Not applicable.

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
