# Peer review of "Does a Little Difference Make a Big Difference? Bovine β-Casein A1 and A2 Variants and Human Health—An Update"

_ijms, 2022, doi:10.3390/ijms232415637_

Round 1
Reviewer 1 Report
In the current study, bovine β-casein A1 and A2 variants related to human health were reviewed. The topic is interesting and well written, however, several issues should be revised in the manuscript.
Comments :
1. Lines 56-57, the abbr. of these terms were not present in the following text, could be removed. The others abbr. should be checked.
2. Line 64, this section is not focused on the “…evolution”, “genetic variants” or others that could be better.
3. Line 101 and others, the styles of references should be revised according to the Journal.
4. Line 176, removed a full stop before the cited references.
5. Line 179. Added a full stop after the word of the figure.
6. Line 210, removed “(ESI/MS)”, it was not found in the text again.
7. Lines 241-243, provide the specific digestion. What’s the fresh and hydrolysed by digestive enzymes bovine milk?
8. Line 294, revised as “mL”. what’s the definition of“conventional milk”, it could be normal milk?
9. Line 482, check the “Dong-Ning et al. [170] and Han et al. [170]”.
10. the styles of several terms in the manuscript should be revised for readable, such as, beta-casein or β-casein or β-CN or CSN2, beta-casomorphin-7 or β-casomorphin-7 or BCM-7. In addition, the abbr. of BCN and BCM are not needed, β-casein and β-casomorphin could be better. Further, provide the specific terms for the “cows A1A2 or A1A1” or “A1 or A2 milk”, as well as the gene of “A1A2 or A1A1 β-casein” or “A1 or A2 β-casein”.
Author Response
Author's Reply to the Review Report (Reviewer 1)
Comments and Suggestions for Authors
In the current study, bovine β-casein A1 and A2 variants related to human health were reviewed. The topic is interesting and well written, however, several issues should be revised in the manuscript.
Comments :
- Lines 56-57, the abbr. of these terms were not present in the following text, could be removed. The others abbr. should be checked.
Answer: It was improved in manuscript file.
- Line 64, this section is not focused on the “…evolution”, “genetic variants” or others that could be better.AC
Answer: It was improved in manuscript file.
- Line 101 and others, the styles of references should be revised according to the Journal.
Answer: It was improved in whole manuscript file. We had two references doubled as separate authors which was not correct [ref. 170].
- Line 176, removed a full stop before the cited references.
Answer: It was improved in manuscript file.
- Line 179. Added a full stop after the word of the figure.
Answer: It was improved in manuscript file.
- Line 210, removed “(ESI/MS)”, it was not found in the text again.
Answer: We would like to leave ESI/MS also as abbreviation, because for some scientists this is the only form that they use in laboratory language.
- Lines 241-243, provide the specific digestion. What’s the fresh and hydrolysed by digestive enzymes bovine milk?
Answer: The paragraph lists fresh milk as non-hydrolysed and not treated with any digestive enzymes. In contrast, hydrolyzed milk was subjected to the action of digestive enzymes, which reflected the fate of food in the digestive tract.
- Line 294, revised as “mL”. what’s the definition of“conventional milk”, it could be normal milk? Answer: “mL” was improved in manuscript file.
- Line 482, check the “Dong-Ning et al. [170] and Han et al. [170]”
Answer: Dong-Ning and Han is name and surname of the one author. We are sorry for interpreting it wrongly. We connect this citation in the main body text and it is visible in references as only one reference as it should be. Thanks to the Reviewer for noticing.
- the styles of several terms in the manuscript should be revised for readable, such as, beta-casein or β-casein or β-CN or CSN2, beta-casomorphin-7 or β-casomorphin-7 or BCM-7. In addition, the abbr. of BCN and BCM are not needed, β-casein and β-casomorphin could be better. Further, provide the specific terms for the “cows A1A2 or A1A1” or “A1 or A2 milk”, as well as the gene of “A1A2 or A1A1 β-casein” or “A1 or A2 β-casein”.
Answer: As suggested by the Reviewer, the style of notions and abbreviations of the gene (CSN2), protein (β-casein; β-CN), and peptide (β-casomorphin-7, βCM-7) nomenclature have been unified in the manuscript.
The authors would like to thank to Reviewer for taking the time reviewing this work. We are grateful for all the comments and suggestions. This gave us the opportunity to improve our publications but also gave a new look to our research. All suggestions are included in our improved manuscript.
Reviewer 2 Report
This is a very interesting review with a high significance for the readers and the scientific community. The authors present all aspects of the subject, as this is necessary. However, as the subject is relatively unknown to the readers, they should state in the abstract, the discussion and the conclusions what their most important conclusion is and what they believe is scientifically sound for the research community to do.
Author Response
Author's Reply to the Review Report (Reviewer 2)
Comments and Suggestions for Authors
This is a very interesting review with a high significance for the readers and the scientific community. The authors present all aspects of the subject, as this is necessary. However, as the subject is relatively unknown to the readers, they should state in the abstract, the discussion and the conclusions what their most important conclusion is and what they believe is scientifically sound for the research community to do.
Answer: The topic on the role of casomorphins in physiological processes and their potential role in the etiology of diseases is of particular importance to us. We want readers to be able to receive information on biologically active peptides in an understandable and clear way. Therefore, we would like to thank you for this valuable comment. As suggested by the Reviewer, we added the necessary information in the text and changed the abstract section.
The authors would like to thank to Reviewer for taking the time reviewing this work. We are grateful for all the comments and suggestions. This gave us the opportunity to improve our publications but also gave a new look to our research. All suggestions are included in our improved manuscript.
Author Response
Author's Reply to the Review Report (Reviewer 3)
The paper represents an interesting point of view on this problematic, there a lot of information and the information are supported by bibliography. In my opinion, the authors did a good job and the manuscript should be accept with little correction (minor revision).
Corrections to follow:
TITLE AND ABSTRACT
The Authors should consider the title's change and improve the abstract. This latter has not much appeal respect to the rest of manuscript
Answer: As suggested by the Reviewer, we changed the abstract section. We would like to leave the title in present form to make it more interesting for readers.
AFFILIATION
Please, check the affiliation, there is a problem with numbering and adjust the graphic style of affiliation(space, recess)
Answer: It was improved in manuscript file. We corrected citation style for MDPI using coding template available on MDPI site for Zotero citations. Years of the mentioned articles were deleted from body text according to MDPI style rules.
GENERAL INFORMATION
1. The Authors should add the table with all abbreviation at the end of manuscript, before the references
Answer: It was improved in manuscript file – we added all abbreviation at the end of manuscript.
The Authors should make a unique paragraph on nervous system problems, current there are three paragraphs (7.6; 7.6.1; 7.6.2).
Answer: It was improved in manuscript file.
Check the figures’ resolution (minimum
Answer: We would like to thank you for your suggestions about figures as they are an important part of our review. Improving the quality of figures will certainly be beneficial for readers of the Journal. The figures were improved in manuscript file.
The authors would like to thank to Reviewer for taking the time reviewing this work. We are grateful for all the comments and suggestions. This gave us the opportunity to improve our publications but also gave a new look to our research. All suggestions are included in our improved manuscript.
Round 2
Reviewer 2 Report
The authors have made the necessary alterations. The manuscript can now be published.